# Exploring the Efficacy of Biocontrol Microbes against the Fungal Pathogen *Botryosphaeria dothidea* JNHT01 Isolated from Fresh Walnut Fruit

**DOI:** 10.3390/foods11223651

**Published:** 2022-11-15

**Authors:** Qiu Qin Zhang, Jie Shi, Pei Yao Shen, Fei Xi, Cheng Yu Qian, Guo Hua Zhang, Hai Jun Zhu, Hong Mei Xiao

**Affiliations:** 1College of Food Science and Technology, Nanjing Agricultural University, Nanjing 210095, Chinaxhm@njau.edu.cn (H.M.X.); 2Shandong Wukangxuan Modern Agriculture and Forestry Development Co., Ltd., Zoucheng 273519, China; 3Jiangsu Academy of Agricultural Sciences, Nanjing 210014, China

**Keywords:** walnut, *Botryosphaeria dothidea*, postharvest disease, biocontrol agents, *Bacillus amyloliquefaciens*, *Hanseniaspora uvarum*

## Abstract

Biological control by antagonistic microorganisms are an effective and environmentally friendly approach in postharvest disease management. In order to develop a biocontrol agent for fresh walnut fruit preservation, the potential biocontrol effects of *Bacillus amyloliquefaciens* RD.006 and *Hanseniaspora uvarum* FA.006 against the main fungal pathogen of walnuts were evaluated. *Botryosphaeria* species showed the highest detection, and the JNHT01 strain showed the strongest pathogenicity. *Bot. dothidea* JNHT01 caused gray mold and brown rot on fresh walnuts, and its incidence rate reached 100% after an 8 days incubation. The growth of this fungal strain can be promoted by lighting, with a maximum growth rate achieved at a pH of 7 and at 28 °C. *B. amyloliquefaciens* RD.006 and *H. uvarum* FA.006 supernatants at a concentration of 1–15% *v*/*v* showed antifungal activity. The mycelial growth inhibition rates of *Bot. dothidea* JNHT01 were 23.67–82.61% for *B. amyloliquefaciens* RD.006 and 1.45–21.74% for *H. uvarum* FA.006. During *Bot. dothidea* JNHT01 growth, the biomass, nucleic acid leakage, and malondialdehyde content gradually increased, while the DPPH scavenging capacity and SOD activity decreased. The *B. amyloliquefaciens* RD.006 and *H. uvarum* FA.006 strains showed antifungal activity by damaging fungal cell membranes and reducing fungal antioxidant activity. Moreover, the antifungal effect of *B. amyloliquefaciens* RD.006 was higher than that of *H. uvarum* FA.006. Hence, the RD.006 strain of *B. amyloliquefaciens* can be considered a potential biocontrol agent for the management of postharvest walnut diseases caused by *Bot. dothidea*.

## 1. Introduction

Postharvest losses that occur during the transport of harvested fruits and vegetables from the field to the farm, handling, and storage are considerable. Food and Agriculture Organization (FAO) estimates that postharvest losses can reach up to 20 percent for fruit and vegetables in the world [1]. Postharvest losses are typically caused by postharvest diseases, which result in a reduction in flavor, texture, color, nutrition, and quantity [2]. The majority of fruit pathogenic microorganisms are fungi, such as *Alternaria* species that cause sooty mold, *Botrytis* species that cause gray mold, *Penicillium* species that cause blue mold, *Colletotrichum* species that cause bitter rot, *Rhizopus* species that cause soft rot [3], etc. Most spoilage fungi naturally grow in suitable environmental conditions during pre- and post-harvest periods, while certain types of fungi are usually produced by mycotoxin. Fungi infection and mycotoxin contamination cause fruit decay and poisoning [4]. Given the significant financial, nutritional, and environmental impacts of postharvest losses for both farmers and consumers, postharvest disease management is a key challenge in food supply chains and is of high importance for fruit security and income generation.

Walnuts (*Juglans regia*) are a nutritional food for promoting cardiovascular, neurological, and immune system health due to their high content of unsaturated fatty acids, minerals, vitamins, and photochemicals. Fresh walnuts are highly perishable and difficult to preserve [5,6]. One of the main factors involved in postharvest decay of walnut is mold fungi infection in the field or the postharvest supply chain [7]. *Botryosphaeria dothidea*, a species of *Botryosphaeria*, is known to cause canker- and dieback-related diseases. This fungal pathogen is widely distributed in fruits, including walnuts [8], blueberries [9], apples [10], and kiwifruits [11]. Branch dieback, brown blight, and canker diseases caused by *Bot. dothidea* were widely observed in walnut trees all over the world, including China [8], Turkey [12], South Korea [13], Japan [14], and the USA [15], etc. In order to effectively reduce postharvest decay and extend the storage time, preservation technologies should be proposed to inactivate mold fungi, especially *Bot. dothidea*, on walnuts.

Various preservation methods, including radiation [16], edible coating [17], low-temperature storage [5], and essential oil vapors [18], have been developed to prolong the shelf life of walnuts. However, antifungal therapy with these approaches has had few positive effects. Recently, a variety of antagonistic microorganisms have be discovered and applied to harvested produce as biocontrol agents to direct against postharvest diseases [19]. In our previous study, *H. uvarum* demonstrated relatively high levels of efficacy against postharvest mold decay of strawberries [20]. The postharvest phase has been considered a very suitable environment for the successful application of biocontrol agents to extend the shelf life of fruits. Application of biological agents is practical and can be effective in the reduction of agrochemical use and residues. Hence, biological control by microbial antagonists is a sustainable, energy-efficient, and environmentally friendly approach to the management of postharvest diseases.

Biocontrol agents can be effective for the management of commercial postharvest diseases. Although most walnut cultivars are susceptible to infection with *Bot. dothidea,* biological control for the management of walnut diseases is limited. We obtained two strains, *Bacillus amyloliquefaciens* RD.006 and *Hanseniaspora uvarum* FA.006, which have potential biocontrol effects against postharvest diseases. The objective of this study was to identify the main fungal pathogen of walnuts and evaluate the biocontrol potential of *B. amyloliquefaciens* RD.006 and *H. uvarum* FA.00 against the main pathogenic fungus.

## 2. Materials and Methods

### 2.1. Fruit Sample and Antagonistic Agent

Green walnut (*Juglans regia* L. Liaoning 1) fruits with disease symptoms were collected during 2021–2022 from Wukangxuan orchard, Zoucheng City, Shandong Province, China, and delivered to the laboratory within 48 h.

*B. amyloliquefaciens* RD.006 and *H. uvarum* FA.00 used in the present study were isolated from marine soil and healthy strawberry fruit, respectively. These two strains were obtained from the College of Food Science and Technology, Nanjing Agricultural University, China. 

### 2.2. Isolation and Purification of Pathogenic Fungi

Pathogenic fungi were isolated from diseased fruit samples by the tissue block separation method based on Koch’s rule [21]. Tissue of samples with disease symptoms were cut into 3 mm × 4 mm segments with a sterile knife, surface-disinfested with 75% alcohol for 10 s, rinsed 3 times with sterile distilled water, and dried with sterile filter paper. Three segments of each sample were placed on a potato dextrose agar (PDA) plate in the shape of a triangle and incubated at 28 °C in a constant temperature incubator for 3 days. Pure cultures were obtained by sub-culturing from the grown hyphal tips.

### 2.3. Identification of Pathogenic Fungi

#### 2.3.1. Morphological Identification

The purified strains were inoculated on PDA and cultured at 28 °C with 75% relative humidity for 5 days. The characteristics of colonies were observed, and the morphology of mycelia and spores was observed under a microscope.

#### 2.3.2. Molecular Identification 

After mycelium growth, DNA was extracted with the Genomic DNA Extraction Kit (Tiangen, China). PCR amplification was carried out with ITS_1_ (TCCGTAGGTGAACCTGCGG) and ITS_4_ (TCCTCCGCTTATTGATATGC) primers for ITS-rDNA; NS1 (GTAGTCATATGCTTGTCTC) and NS8 (TCCGCAGGTTCACCTACGGA) for 18s rDNA. The total volume of the PCR amplification system was 25 μL containing 0.5 μL template DNA, 2.5 μL Mg^2+^ buffer, 1 μL dNTP, 0.2 μL enzyme, and 1.0 μL primers. The amplification of genes was carried out with the following thermal cycling profile: an initial denaturation at 94 °C–98 °C for 4–5 min, 30 cycles of 94 °C–98 °C for 30–45 s by, 30 cycles of 55 °C for 30–45 s, ending with a final extension at 72 °C for 10 min. DNA sequence data was analyzed by Shanghai Shenggong Biotechnology Co., Ltd. (Shanghai, China) and BLASTed in the GenBank database. The evolution tree was drawn by mega-x software according to the adjacency method.

### 2.4. Characterization of the Fungal Pathogen

#### 2.4.1. Pathogenicity of the Fungal Pathogen on Walnuts

The identified fungal mycelium was used for the pathogenicity evaluation. One eight mm hole was made in the equatorial surface of the walnut sample using a sterilized hole punch. The fungi were cultured on PDA medium for 2–3 days and cut into a 5 mm-diameter mycelium blocks when the mycelium was about 1 mm thick. For pathogenicity experiments, mycelium blocks were transferred into the hole and incubated at 28 °C and RH 90–95% for 8 days. The control was set under the same conditions without adding any pathogenic fungi. The symptoms and diameter of the spots were observed.

The fruit disease index was calculated:(1)$$D=\frac{\sum (N_{1}\times A)}{N_{0}\times \;B}\times 100\%$$
where D is the disease index; N_1_ is the number of diseased fruits; N_0_ is the total number of inoculated fruits; A is the representative number; B represents the numerical value of the most affected fruit.

The incidence rate of pathogenic fungi was calculated:(2)$$M=\frac{N_{1}}{N_{0}}\times 100\%$$
where M is the incidence rate (%); N_1_ is the number of diseased fruits; N_0_ is the total number of inoculated fruits.

#### 2.4.2. Growth Characteristics of Pathogenic Fungi

To detect the effect of light on fungi growth, PDA plates with pathogenic fungi were incubated under three light conditions (24 h dark, 12 h light followed by 12 h dark, 24 h light per day) at 28 °C and 75% RH. To detect the effect of temperature on fungi growth, PDA plates with pathogenic fungi were incubated at 15 °C, 20 °C, 25 °C, 28 °C, 30 °C, 35 °C, and 40 °C under a dark condition for 24 h per day and 75% RH. To detect the effect of pH on fungi growth, pathogenic fungi were inoculated to PDA cultures with amended pH values (5, 6, 7, 8, and 9) and incubated in the dark for 24 h at 28 °C and 75% RH. After 2 days of inoculation, the diameter of the colony was measured.

### 2.5. Determination of Antifungal Potential of Biocontrol Agents

#### 2.5.1. Preparation of Cell-Free Supernatants

*B. amyloliquefaciens* RD.006 and *H. uvarum* FA.006 were respectively cultured in Luria-Bertani and Potato Dextrose Broth medium at 28 °C for 14 h at 150 rpm reached an OD600 of approximately 2.0. The fermentations were centrifuged for 10 min at 8000 r min^−1^ at 4 °C, and then were filtered through a pre-sterilized 0.22 um membrane filter to collect cell-free supernatants.

#### 2.5.2. Antifungal Concentration of Biocontrol Agents

The culture medium PDA with cell-free supernatant at different concentrations (1%, 2.5%, 5%, 10%, and 15%) was poured into petri plates. One 5 mm hole was made at the center of the PDA plate using a sterilized punch. Pathogenic fungal mycelium was added into the hole and then incubated at 28 °C for 5 days to measure the colony diameter. Inhibition rate = (D_0_ − D)/D × 100%. Where, D_0_ is the colony diameter on the PDA without cell-free supernatant (mm) and D is the colony diameter on the PDA with cell-free supernatant (mm).

#### 2.5.3. In Vitro Antagonistic Activity Determination

Measurement of the antagonistic activities of *B. amyloliquefaciens* RD.006 and *H. uvarum* FA.006 against the main pathogens were conducted with in vitro experiments. About 0.1 g of pathogenic fungi were inoculated into 160 mL of PDB culture solution and incubated at 28 °C for 24 h at 150 rpm. Then the cell-free supernatants were added into cultures to a final concentration of 10% and continued to incubate for 8, 16, or 24 h. Sterile water was added as the control group. The cultures were either filtered through four layers of pre-sterilized gauze or centrifuged at 6000 r min^−1^ for 15 min to separate the supernatant and hyphae of pathogenic fungi. The biomass, nucleic acid leakage, malondialdehyde content, DPPH (2,2-Diphenyl-1-picrylhydrazyl) scavenging capacity, and SOD activity were measured.

The mycelium was washed with deionized water and then dried at 60 °C to a constant weight. The dried mycelium was weighed and recorded as biomass.

Nucleic acids in the supernatant were measured through absorbance measurements at a wavelength of 260 nm by a UV-Vis spectrophotometer according to the methods described by Cai et al. [22].

Malondialdehyde was determined by the 2-thiobarbituric acid method according to the methods described by Xiong et al. [23].

DPPH scavenging ability was determined according to the methods described by Li et al. [24].

SOD activity was determined by assaying the ability to inhibit the photochemical reduction of nitroblue tetrazolium chloride according to the methods described by Zhang et al. [25].

### 2.6. Data Statistics and Analysis

Each experiment was done in triplicate. The data values were expressed as means ± standard deviation. Statistical analysis was performed with a one-way analysis of variance followed by Duncan’s multiple range test using the SPSS 16.0 software (SPSS Inc., Chicago, IL, USA). Differences at *p* < 0.05 were considered statistically significant.

## 3. Results

A total of 10 fungi were isolated from mold fresh walnuts according to Koch’s rule. These isolated fungi are strains with typical symptoms that cause the mildew, blackening, and decay of fresh walnuts (Figure 1). Except for strains HT05, HT06, and HT10, other isolates cause black rot in fresh walnuts. Figure 1 also showed that the isolate strain JNHT01 showed the strongest pathogenicity among the 10 isolates. The inoculated hulls of strain JNHT01 were black and decayed, with a large number of hyphae around the wound sites, indicating that this strain caused gray mold and brown rot in fresh walnuts.

The highly conserved ITS-rDNA or 18S rDNA regions were sequenced for these seven strains which caused brown blight of walnuts. A homology search with the BLASTN program at NCBI showed that the sequences from the isolates showed 99–100% similarity to the reference sequences (Table 1). These strains were identified as *Botryosphaeria dothidea*, *Fusarium proliferatum*, *Colletotrichum siamense*, and *Colletotrichum gloeosporioides*. *Bot. dothidea* had the highest detection rate, with three out of seven strains. 

Figure 2A shows the characteristics of the hypha of the pathogenic isolate *Bot. dothidea* JNHT01 of walnut. This JNHT01 strain, grew well on PDA at 28 °C for 5 days under these ecophysiological conditions. A homology search with the BLASTN program at NCBI showed that the ITS sequences from the isolates showed almost 87% similarity to the reference sequences of *Bot. dothidea* JZG1 (Figure 2B).

In order to confirm the pathogenicity of *Bot. dothidea* JNHT01, the diameter of the disease spot, the disease index, and the incidence rate of walnut fruit were evaluated at 5 and 8 days incubation. The fruit was easily decayed, and the disease indexes were 22.22% at 5 days and 72.22% at 8 days. The incidence rate of *Bot. dothidea* JNHT01 was 66.67% with a 0.80-mm lesion diameter after 5 days incubation and was 100% with a 3.00 mm lesion diameter after 8 days incubation (Table 2). Biological control of *Bot. dothidea* JNHT01 was carried out in the next study due to its high pathogenicity.

Figure 3 illustrates the effects of the environment, including light time, pH, and temperature, on mycelium growth. The mycelium of *Bot. dothidea* JNHT01 has a significantly (*p* < 0.05) higher rate in the full light than that in the light shading and the dark (Figure 3A), indicating that light illumination may enhance the fungal infection of *Bot. dothidea* JNHT01 on walnut. The possible responses based on the effects of pH (ranging from 5 to 9) and temperature (ranging from 10 °C to 40 °C) were investigated (Figure 3B,C). The growth of *Bot. dothidea* JNHT01 was pH- and temperature-dependent, and the highest mycelium was obtained at pH 7 and 28 °C.

In this study, *B. amyloliquefaciens* RD.006 and *H. uvarum* FA.006 were applied as biocontrol agents. After 14 hours of incubation, these two strains were in stationary phases and used for biological control experiments. To understand their inhibitory efficacy against *Bot. dothidea* JNHT01, different concentrations (ranging from 1% to 15% *v*/*v*) of cell-free culture supernatants were added to PDA to evaluate the antifungal activities (Figure 4). *Bot. dothidea* JNHT01 was significantly (*p* < 0.05) inhibited by these two cell-free culture supernatants. The inhibition rate by *B. amyloliquefaciens* RD.006 was 23.67–82.61%, while *H. uvarum* FA.006 was 1.45–21.74%. The inhibitory activity of *B. amyloliquefaciens* RD.006 was at least 3.7 times higher than that of *H. uvarum* FA.006. The inhibition of biocontrol agents showed a progressive increase with concentration. There was no significant (*p* > 0.05) difference in inhibitory efficacy at 10% and 15%. In consideration of the inhibitory efficacy, culture supernatants at 10% were selected as biocontrol agents for an in vitro antifungal experiment.

Figure 5 illustrates the inhibitory effects of biocontrol agents at 10% concentration on the growth of *Bot. dothidea* JNHT01. After 24 h of incubation, the biomass of *Bot. dothidea* JNHT01 reached 717.45 mg. There was no significant (*p* > 0.05) *Bot. dothidea* JNHT01 growth reduction between *B. amyloliquefaciens* RD.006 and *H. uvarum* FA.006 over 16 h of incubation, but it showed significant (*p* < 0.05) inhibition at 24 h.

During incubation, the nucleic acid leakage and malondialdehyde content of *Bot. dothidea* JNHT01 gradually increased, which was enhanced after 10% biocontrol agent treatment (Figure 6). Compared to the untreated group, the nucleic acid leakage and malondialdehyde content of *B. amyloliquefaciens* RD.006 treatment were significantly (*p* < 0.05) high over 24 h of incubation, while *H. uvarum* FA.006 treatment was not significantly different (*p* > 0.05) during 0–16 h of incubation.

The DPPH scavenging capacity and SOD activity of *Bot. dothidea* JNHT01 gradually decreased from 0.24 to 0.14 mol g^−1^ min^−1^ and from 8.71 to 1.65 U mg^−1^ min^−1^, respectively (Figure 7), which indicate the biocontrol agent treatments would lead to a faster reduction of ROS scavenging capacity. *B. amyloliquefaciens* RD.006 treatment obtained the lowest values of DPPH scavenging capacity and SOD activity. Compared to the untreated group, *B. amyloliquefaciens* RD.006 significantly decreased more than 0.05 mol g^−1^ min^−1^ for DPPH scavenging capacity and 1.04 U mg^−1^ min^−1^ for SOD activity. In contrast, *H. uvarum* FA.006 showed significantly lower (*p* < 0.05) DPPH scavenging capacity at 24 h and significantly lower (*p* < 0.05) SOD activity at 16 h and 24 h compared to the untreated group.

Pearson’s correlation coefficients between indicators are shown in Figure 8. Biomass displayed a positive correlation with nucleic acid leakage but showed a negative correlation with malondialdehyde content, DPPH, and SOD activity. At the 0.05 level, DPPH displayed a significant negative correlation with nucleic acid leakage and a significant positive correlation with malondialdehyde content. There were no significant differences among the other indicators. This result suggests that the antifungal activities of biocontrol agents could be attributed to their ability to damage fungal cells and the antioxidant system, especially nucleic acid leakage in *Bot. dothidea* JNHT01.

## 4. Discussion

In this study, *Bot. dothidea*, *F. proliferatum, C. siamense,* and *C. gloeosporioides,* which caused gray mold and brown rot in fresh walnut fruits, have been isolated. Pinter et al. reported [26] that in Hungary, walnut trees were mainly infected by *Phomopsis*, *Diplodia*, *Cytospora,* and *Phoma* spp., while walnuts were primarily infected by *Xanthomonas* and *Gnomonia* spp. during storage. Nkwonta et al. [27] found that *Aspergillus*, *Fusarium*, and *Penicillium* spp. were the main potentially mycotoxigenic fungal groups frequently isolated from African walnuts. The dominant species of walnut in Northwestern China were *Fusarium, Alternaria*, and *Penicillium* spp. [28]. The different between fungal contaminations of walnut is due to the influence of local environment. *Among all isolates, Botryosphaeria* species showed the highest detection rate and pathogenicity to cause black spots. *Bot. dothidea* is also the principal causal agent of other postharvest fruit diseases, such as ring rot of apples [29], soft rot of kiwifruits [30], and black spot of pecans [31]. This research confirms that *Bot. dothidea* is one of the main pathogens for controlling postharvest diseases of walnut. This species frequently infects wounds on young almond and mature walnut and can be found year-round on diseased branches [15]. Hence, this fungus has been isolated from a large number of blighted shoots of walnut and causes walnut canker [8]. Recently, Li et al. [32] observed *Botryosphaeria* infection on English walnut branches in China. Hence, the isolation of *Bot. dothidea* JNHT01 was mainly due to the fungal infection of walnut trees before harvest. Since fruits from *Bot. dothidea* JNHT01 were isolated from walnuts with disease symptoms, fungal contamination of fruits from this strain are mainly due to fungal infection of walnut trees before the harvest.

Chemical fungicides can effectively control *Botryosphaeria* diseases, but they can only be used on fruit trees during the preharvest period. During and after the harvest, *Bot. dothidea* can be controlled by various natural, non-toxic compounds, such as essential oils [33], antibiotics [34], and biocontrol agents [9,35]. In these cases, biocontrol agents are more attractive and effective due to their substitutability of chemicals and broad-spectrum antifungal activity. It has been reported in many publications that biocontrol agents, such as *Meyerozyma guilliermondii* Y-1 [35] and *Streptomyces* sp. CX3 cell-free supernatant [9], can effectively combat pathogen infection and postharvest decay caused by *Botryosphaeria dothidea*. In agreement with previous studies, cell-free culture supernatants from *B. amyloliquefaciens* RD.006 and *H. uvarum FA.00* significantly inhibited the growth of *Bot. dothidea*, indicating that these two biocontrol agents have the potential to suppress mycelial growth, lesion diameter, spore germination, and rot severity of *Bot. dothidea* in walnuts. Antagonistic microorganisms can be used as liquid culture or cell-free culture supernatant on fruits [36]. Both of these two applications exhibit strong antibacterial and antifungal activity against phytopathogens. Wang et al. reported that spraying biocontrol agents before inoculation with a pathogen can achieve better biocontrol efficacy as compared to spraying at the same time or after the fungal inoculation [9]. Hence, cell-free culture supernatant of *B. amyloliquefaciens* RD.006 and *H. uvarum* FA.00 can be used as a sustainable alternative for chemical fungicides and is preferably used before pathogen infection.

Some biocontrol agents have been developed and commercialized against phytopathogens. Two of the major species used as active ingredients are *Bacillus* and *Hanseniaspora* genera. For example, *Bacillus* species have been incorporated in commercial producers of Biosubtilin^®^, Cease^®^, Kodiak^®^, and Avogreen^®^*,* etc. [37,38]. *Hanseniaspora*-based agents also have been reported to be effective against fruit rots [20,39] and mycotoxin contamination [40,41] caused by plant pathogens. For *Bot. dothidea* controlling, numbers studies showed that *B. amyloliquefaciens* exhibited potential values [30,42], but application of H. uvarum agents is rarely been reported. In the current study, the in vitro bioassay results showed that *B. amyloliquefaciens* RD.006 and *H. uvarum* FA.006 were useful in walnut prevention, but the biocontrol efficacy of *B. amyloliquefaciens* RD.006 was higher than that of *H. uvarum* FA.006. Previous studies found that the antagonistic potential of B. amyloliquefaciens was related to lipopeptide activity [43], whereas H. uvarum can produce antifungal volatile organic compounds [20,40,44]. Hence, we hypothesize that the difference in antifungal activity is largely due to the variation of metabolites.

The action of antagonistic microorganisms is associated with several mechanisms, such as competing with pathogens for pace and nutrients, inducing host resistance, and forming formation, et al. [45]. In the present study, during *Bot. dothidea* JNHT01 growth, biocontrol agent treatment promoted the increase of biomass, nucleic acid leakage, and malondialdehyde content. Levels of nucleic acid leakage and malondialdehyde content directly reflect membrane permeability and lipid peroxidation, respectively [10]. This result is in agreement with Wen et al. [42], who reported that an antifungal peptide produced by *B. amyloliquefaciens* could destroy the cell walls of the hyphae of pathogenic fungi. Hamilton et al. [46] found extracellular metabolites of antagonistic microorganisms could damage the fungal cell membranes and lead to electrolyte leakage from the mycelium. The DPPH scavenging capacity and SOD activity of *Bot. dothidea* JNHT01 treated by biocontrol agents were higher than those of the untreated. DPPH scavenging capacity and SOD activity reflect antioxidant activities, which are associated with fungal adaptability to different environmental and stress conditions and host-infection ability [46]. Besides SOD, Shi et al. [47] found that preservative treatment can reduce enzyme activities, including CAT, POD, NADP-malate dehydrogenase, succinate dehydrogenase, cellulase, and glucanase, in *Bot. dothidea*. Hence, the growth-inhibition activity of biocontrol agent against *Bot. dothidea* JNHT01 may be related to the interference on cell membrane and enzyme. Application of biocontrol agents could inhibit mycelial growth *Bot. dothidea* JNHT01 by damaging fungal cell membranes and reducing fungal antioxidant activity.

## 5. Conclusions

*Botryosphaeria dothidea*, *Fusarium proliferatum*, *Colletotrichum siamense*, and *Colletotrichum gloeosporioides* have been isolated from fresh walnut fruits with disease symptoms. *Bot. dothidea* JNHT01 showed the strongest pathogenicity among all isolates and causes gray mold and brown rot in fresh walnuts. The growth of this fungal strain can be promoted by increasing light time, and its maximum growth rate exhibited at pH 7 and 28 °C. *B. amyloliquefaciens* RD.006 and *H. uvarum* FA.006 showed concentration-dependent inhibition in *Bot. dothidea* JNHT01. The biocontrol efficacy of *B. amyloliquefaciens* RD.006 was at least 3.7 times higher than *H. uvarum* FA.006. During the growth of *Bot. dothidea* JNHT01, biomass displayed a positive correlation with nucleic acid leakage but showed a negative correlation with malondialdehyde content, DPPH, and SOD activity. These two microorganisms displayed biocontrol potential against *Bot. dothidea* JNHT01 by inhibiting the increases in biomass, nucleic acid leakage, and malondialdehyde content and the decreases in DPPH and SOD activity during *Bot. dothidea* JNHT01 growth. The present study indicates that postharvest disease of walnuts caused by *Bot. dothidea* JNHT01 can be biologically controlled with *B. amyloliquefaciens* RD.006 and *H. uvarum* FA.006. Moreover, *B. amyloliquefaciens* RD.006, as a biocontrol agent, is more suitable for walnut cultivation and prevention than *H. uvarum* FA.006.

## Figures and Tables

**Figure 1 foods-11-03651-f001:**
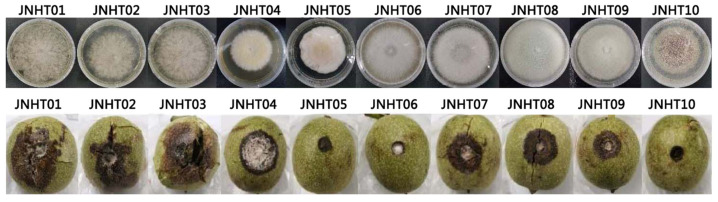
Colony characteristics and pathogenicity of fungal pathogens isolated from fresh walnut.

**Figure 2 foods-11-03651-f002:**
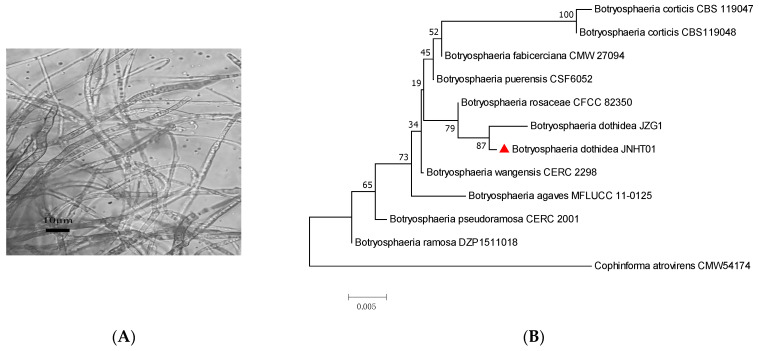
Hypha (**A**) and phylogenetic tree (**B**) of *Bot. dothidea* JNHT01, the fungal pathogen isolated from fresh walnuts. ▲ indicated the isolated strain *Bot. dothidea* JNHT01 from walnut in the present study.

**Figure 3 foods-11-03651-f003:**
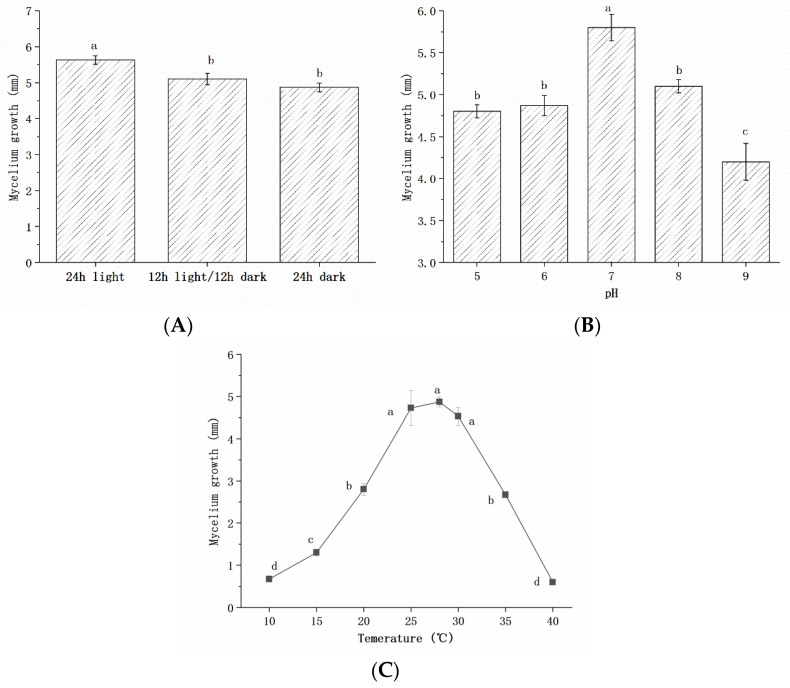
Effects of light illumination (**A**), pH (**B**), and temperature (**C**) on the growth of *Bot. dothidea* JNHT01. Error bars represent standard deviations of the mean. a–d: different letters indicate significant differences with a *p* < 0.05.

**Figure 4 foods-11-03651-f004:**
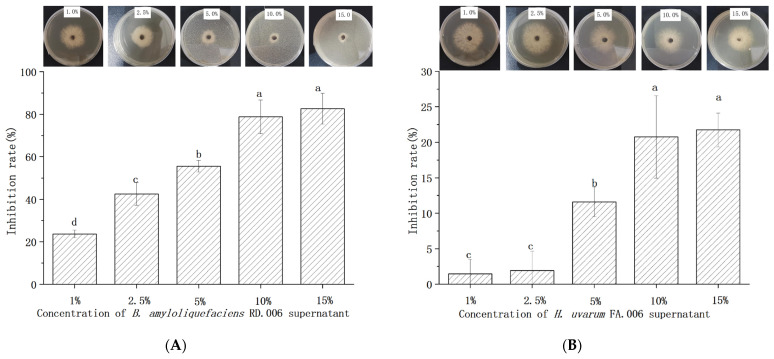
Antifungal activities of cell-free culture supernatants from *B. amyloliquefaciens* RD.006 (**A**) and *H. uvarum* FA.006 (**B**) at different concentrations. Error bars represent standard deviations from the mean. a–d: different letters above bars indicate significant differences at *p* < 0.05.

**Figure 5 foods-11-03651-f005:**
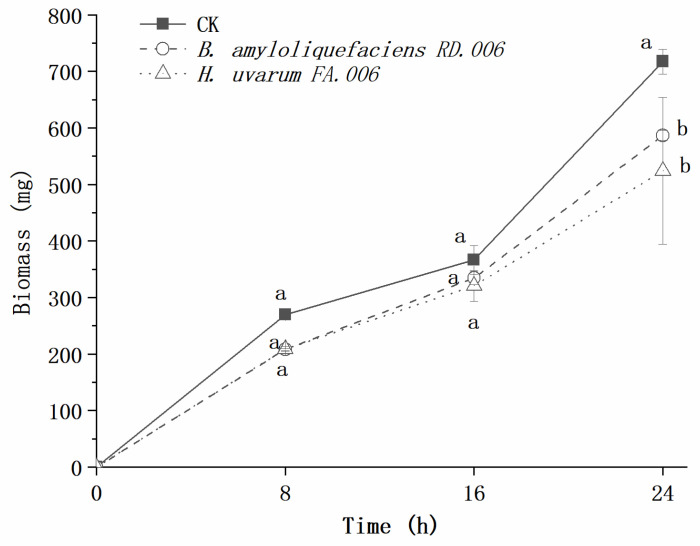
Inhibitory effects of a biocontrol agent at 10% concentration on the growth of *Bot. dothidea* JNHT01. Error bars represent standard deviations from the mean. a–b: different letters at the same time indicate significant differences with a *p* < 0.05.

**Figure 6 foods-11-03651-f006:**
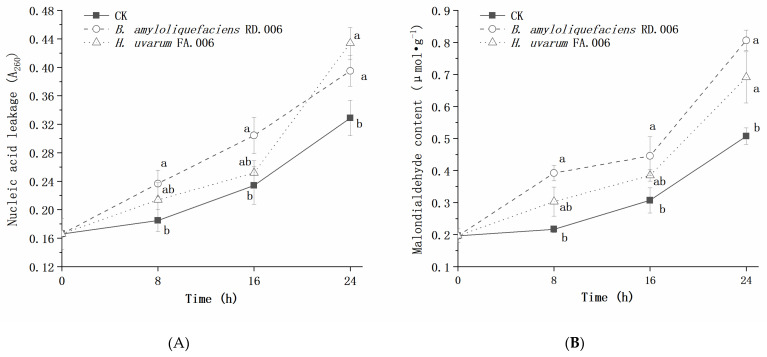
Effect of a biocontrol agent on nucleic acid leakage (**A**) and malondialdehyde content (**B**) of *Bot. dothidea* JNHT01. Error bars represent standard deviations from the mean. a–b: different letters at the same time indicate significant differences with a *p* < 0.05.

**Figure 7 foods-11-03651-f007:**
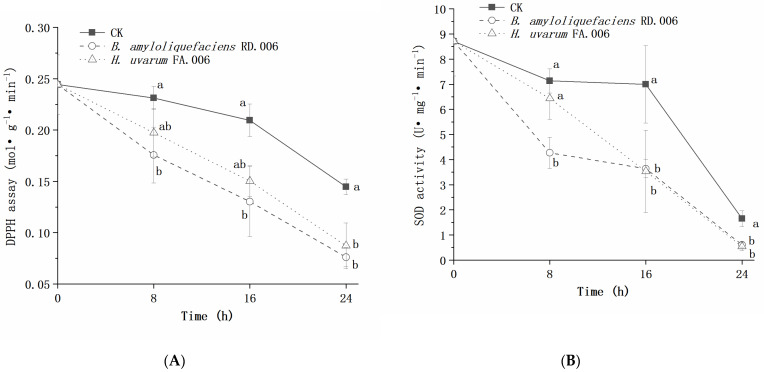
Effect of a biocontrol agent on DPPH scavenging capacity (**A**) and SOD activity (**B**) of *Bot. dothidea* JNHT01. Error bars represent standard deviations from the mean. a–b: different letters at the same time indicate significantly differences with a *p* < 0.05.

**Figure 8 foods-11-03651-f008:**
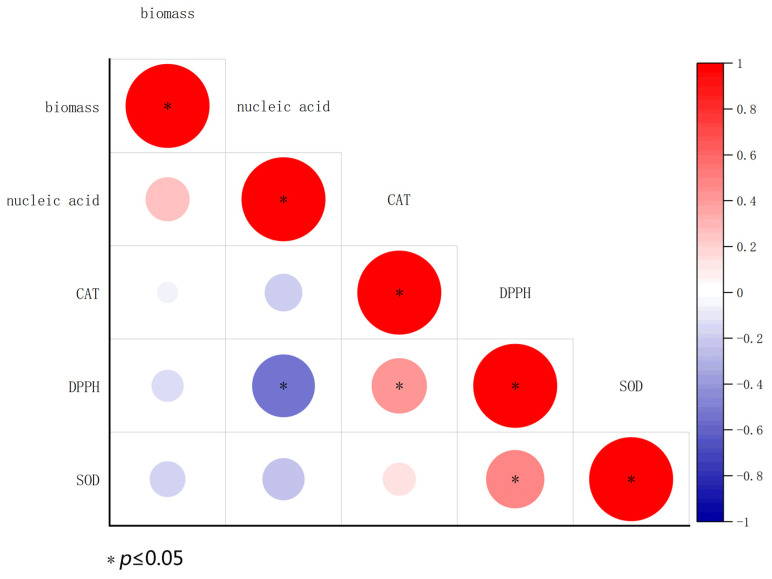
Correlation (Spearman’s) among indicators of *Bot. dothidea* JNHT01 after biocontrol agent treatment. The asterisks indicate significant differences with a *p* < 0.05.

**Table 1 foods-11-03651-t001:** Identification results of fungal pathogens.

Number of Isolates	Accession Numbers of Reference Strain	Sequence Similarity (%)	Identified Species	Accession Numbers of Isolates
JNHT01	MT177976.1	100.00	*Botryosphaeria dothidea*	OP688105
JNHT02	MT177976.1	100.00	*Botryosphaeria dothidea*	*OP741034*
JNHT03	MT177976.1	100.00	*Botryosphaeria dothidea*	*OP741050*
JNHT04	OL614486.1	99.61	*Fusarium verticillioides*	OP740381
JNHT07	ON329281.1	99.27	*Colletotrichum aenigma*	OP740385
JNHT08	KP900277.1	99.44	*Colletotrichum gloeosporioides*	OP740386
JNHT09	MW228103.1	99.26	*Colletotrichum siamense*	OP740387

**Table 2 foods-11-03651-t002:** The pathogenicity of *Bot. dothidea* JNHT01 on fresh walnuts at 5- and 8-day incubation.

Incubation Time (d)	Diameter of Disease Spot (cm)	Disease Index (%)	Incidence Rate (%)
5	0.80 ± 0.25	22.22 ± 3.40	66.67
8	3.00 ± 0.24	72.22 ± 3.40	100

## Data Availability

The data presented in this study are available on request from the corresponding author.

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
