# Peer review of "Exploring the Efficacy of Biocontrol Microbes against the Fungal Pathogen Botryosphaeria dothidea JNHT01 Isolated from Fresh Walnut Fruit"

_foods, 2022, doi:10.3390/foods11223651_

Round 1
Reviewer 1 Report
The manuscript FOODS-1984973 by Qiuqin Zhang reports mainly on the effects of a bacterial and a yeast strain as biocontrol agents against a strain of Botryosphaeria dothidea isolated from fresh walnut. The topic is of interest from a practical point of view due to the possibility to find alternatives to chemicals to manage fruit diseases. However, it cannot be accepted for publication in its present form and needs strong revision before being considered for re-submission.
First of all it must be checked and corrected by an English speaking person since it contains several grammatical errors, mistakes, typos, incomplete and senseless sentences which make it hardly readable and somewhere fully comprehensible.
Following are the main shortcomings:
- As one of the aims of the study was the identification of the main fungal pathogen of walnuts, the main issue is that a too low number of strains (n=10) has been isolated to be representative of the fungal population responsible for postharvest disease. This is a big weakeness as such limited number is not significant at all. Moreover, concerning the first part of this work (i.e. the isolation, identification and characterization of fungal isolates) I do not agree with the authors’ decision to skip description of the corresponding results as such experimental phase resulted in the selection of one out of the 10 isolates, which was then used for further analysis. This choice is/seems to be absolutely arbitrary and not supported by any robust experimental data.
- Based also on these comments and criticisms, authors should focus the manuscript on the characterization of the strain HT01 and avoid referring to Botryosphaeria dothidea/main fungal pathogen in general. This must be revised through the whole manuscript and the title corrected by including the strain HT01 too.
- The induction should be strongly implemented as it is rather general with basic contents, while more detailed information and references to updated studies should be included. Just as an example, no report on the relevance and incidence of Botryosphaeria dothidea is presented. Similarly, discussion is overall really weak and should be improved by referring also to the most updated studies on Botryosphaeria dothidea, Bacillus amyloliquefaciens and Hanseniaspora uvarum as biocontrol agents, and their mechanism of action. Overall, the authors should highlight and comment the scientific reason behind the results which are currently missing.
- There are some imprecisions and inconsistencies between through the manuscript which must be corrected both in the abstract, results and discussion section and the conclusions which, otherwise, are not totally supported by data.
- Concerning the experimental plan, it is not clear how authors standardised the inoculum size and which inoculum rates were in fact used . This should be better described. In this view it is also unclear why mycelia instead f spores were used for the various tests.
Below are some specific comments.
Reference n° 1 is a bit old and it should be replaced by a more recent one.
L 35-37 Please change the sentence as “… nutrition and quantity, and safety issues related to the production of toxic compounds”
L 37 Please delete “pathogenic”
L 37 – 40 Please rephrase the sentence as it is grammatically incorrect, and its end is too vague, i.e. “et al” does not give any information to the readers.
L 40-42 Please correct the sentence as “ .. while certain types of fungi usually produce mycotoxin during the pre-harvest periods.”
L 43-46 Please correct the sentence as “ and highly important for fruits security and income generation”.
L 47-51 The description provided into brackets for chemical preservatives does not provide any additional information for readers, but it is simply a repetition of the idea of using chemicals. Authors should change it by detailing the main content.
L 54-56 Please delete the sentence as it is scientifically and practically incorrect. In its present form, the message delivered to readers is that “yeasts or bacteria used/proposed as biocontrol agents against fruit disease agent are considered harmless to humans”, also considering that biological agents undergo the Qualified Presumption of Safety which was developed to provide a pre-evaluation of the safety by EFSA. Moreover, the list of QPS recommended biological agents has recently been updated by EFSA in 2021.
L 56-57. Please correct “have” into “has”.
L 57 -60 Please correct the sentence as it contains some errors (e.g. “pace”; should it be “Space”?), unclear/missing parts (“and forming formation”) and its end is too vague, i.e. “et al” does not give any information to the readers.
L 61 -63 Please correct as “Application of biological agent can effectively contribute to…”
L 71 Please correct “In order”
Materials and Methods
L 86-87. Please indicate when walnuts were collected, how long and how, i.e. under which conditions they were stored before experiments were carried out.
L 86-87 Is “wukangxuan” a place or what else? Please clarify and detail better for clarity to the readers.
L 120-129 Authors should better detail how experiments were standardised and carried out. For example, no information is provided for the cultivation conditions nor for the standardization of the inocula: how authors can assure they used the same inoculum rates for the various fruits since “mycelium blocks” were used to inoculate the fruits?
Also, how many walnut fruits were used to test the pathogenicity of the fungal strains on walnuts? Formula should be better explained by detailing what they differ for, what the representative number correspond to, and the difference between number of fruit disease and total fruit of disease.
L 131-134 It is unclear what “PDA plants” correspond to. Please clarify by adding more details to this section.
L 132-134 Please add also 40°C to harmonise with results presented in figure 3c.
L 131-137 According to L 137, assessment of the effects of different light/dark conditions, different temperature and pH values was made 2 days after inoculation. However, the incubation time is reported to be 24 hours. Such an inconsistency must be corrected.
L 139 I suggest authors to change the heading of the section 2.5 as no evaluation of the presence of proteins was in fact made.
Similarly, the heading of the section 2.5.2 is incorrect as it does not reflect the content of the section which in fact refers to the effects of different contents of supernatants added to agarized medium on the growth extent of the fungi.
L 140-143 Authors should include an explanation for the choice to cultivate the Bacillus and Hanseniaspora strains for 14 hours as such a cultivation time is rather short, especially for the yeast. Also additional information on their inoculum rates and final cell loads after their growth, and incubation temperatures as well, should be included.
L 153-173 According to the description of the section 2.5.3, no cell free supernatant of the Bacillus/Hanseniaspora cultures was added to the liquid medium during the cultivation of the fungal strains. It is therefore unclear how antifungal activity was evaluated. Also the reference to Ye et al. [18] is not relevant since a different test was used by Ye et al compared to the present work to check for the antifungal activity (i.e. measurements of the inhibition zones on PDA plates by Ye et al. vs weight of the dried biomass following cultivation in liquid medium in the current study).
Moreover, it is important that authors include key information related to the preparation of the inoculum for the fungal strains and the corresponding inoculum rates.
Results
Concerning the first part of this work, i.e. the isolation and characterization of fungal isolates, I do not agree with the authors’ decision to skip description of the corresponding results. Authors should therefore include the accession numbers of the isolates and the corresponding genus and species. These results are extremely relevant also considering that one of the stated aims of the work was to identify the main fungal pathogen of walnuts.
Moreover, also the results of the pathogenicity tests of the fungal isolates on walnuts must be presented since this part of the experimental work is an important phase which resulted in the selection of the strongest pathogen.
L 197-199 This part is incorrect and inconsistent with the description given in the M&M (section 2.4.1). First of all it is unclear why bacterial cell suspension was inoculated into fresh walnuts to confirm the pathogenicity of this isolate. Secondly, according to the M&M fungal mycelium was used in this test, while spore solution is reported to be used at L 198-199.
Table 1 – Is it correct that data recorded for the disease are expressed in mm? It seems to me that 0.8 mm is really a low value; how could authors have measured such values? Also data of the control uninoculated samples should be included in the table. Simply reporting that “The fruit inoculated with spore solution was more easily decayed than the control” (L 198-199) is rather vague for readers.
Figure 3a; L 208-211 Please harmonise labelling of light conditions used for cultivation in fig. 3a with the description of the M&M section. In fact 24-h cultivation under dark is significantly different from heavy shading, and similarly 12 h light followed by 12 h dark is not the same as light shading. Such inconsistency must be removed and authors must clarify which conditions were in fact used since results showed that light conditions strongly affect the extent of fungal growth
L 208-209 The sentence must be revised as the verb is missing.
L 219-220 The word “Different” must be written in lowercase letter.
L 245 Please correct “insignificant” as “not significantly different”
L 258-260 Please correct “significance” into “significantly”
Please correct “untreatment” through the whole manuscript with “untreated” or “control”
Figure 8 and the corresponding text at L 267-274 are not necessary as the do not provide any additional information than those in figures 4-7, and therefore they should be deleted.
L 282-284 – Please edit the sentence as follows “This result suggests that antifungal activities of biocontrol agents could be attributed to their ability to damage fungal cells and antioxidant system, especially nucleic acid leakage of Bot. dothidea HT01”
L 295 -296 Please correct as “ … found that Aspergillus, Fusarium, …………..frequently isolated from African walnuts.”
L 307 Please correct as “biocontrol agents”
L 310-312 See my previous comments on “et al”.
L 320-322 The sentence should be corrected as in fact the biocontrol agents did not inhibit, but promoted the nucleic acid leakage and malondialdehyde content.
Author Response
Response to editor and reviewers
Dear Editor and Reviewers:
We thank you very much for your comments for our manuscript entitled “Exploring the efficacy of biocontrol bacteria against the fungal pathogen Botryosphaeria dothidea JNHT01 isolated from fresh walnut”. We have studied reviewer’s comments carefully and have made revision which marked in red in the revised manuscript. We invited a professional organization to revise the language of the article. We appreciate for Editors/Reviews’ warm work earnestly, and hope that the correction will meet with approval.
Looking forward to hearing from you.
Thank you and best regards.
Yours sincerely.
Hongmei Xiao
Reviewer #1:
First of all it must be checked and corrected by an English speaking person since it contains several grammatical errors, mistakes, typos, incomplete and senseless sentences which make it hardly readable and somewhere fully comprehensible.
Following are the main shortcomings:
As one of the aims of the study was the identification of the main fungal pathogen of walnuts, the main issue is that a too low number of strains (n=10) has been isolated to be representative of the fungal population responsible for postharvest disease. This is a big weakeness as such limited number is not significant at all. Moreover, concerning the first part of this work (i.e. the isolation, identification and characterization of fungal isolates) I do not agree with the authors’ decision to skip description of the corresponding results as such experimental phase resulted in the selection of one out of the 10 isolates, which was then used for further analysis. This choice is/seems to be absolutely arbitrary and not supported by any robust experimental data.
Response: We revised through the whole manuscript. We collected fresh walnut samples from the same source in one season, so the fungal pathogens of walnut infection are relatively consistent. More, we only isolated fungal pathogens from these samples that are moldy during storage. Therefore, the isolated fungi are strains with typical symptoms screened according to Koch's rule. Hence, only 10 isolates were obtained. The colony characteristics and pathogenicity of these ten fungal pathogens were added in figure 1.
Based also on these comments and criticisms, authors should focus the manuscript on the characterization of the strain HT01 and avoid referring to Botryosphaeria dothidea/main fungal pathogen in general. This must be revised through the whole manuscript and the title corrected by including the strain HT01 too.
Response: We revised through the whole manuscript, modified results and discussion. Since there is a Botryosphaeria dothidea strain named HT01 in NCBI, we change our strain to JNHT01
The induction should be strongly implemented as it is rather general with basic contents, while more detailed information and references to updated studies should be included. Just as an example, no report on the relevance and incidence of Botryosphaeria dothidea is presented. Similarly, discussion is overall really weak and should be improved by referring also to the most updated studies on Botryosphaeria dothidea, Bacillus amyloliquefaciens and Hanseniaspora uvarum as biocontrol agents, and their mechanism of action. Overall, the authors should highlight and comment the scientific reason behind the results which are currently missing.
Response: Information about the distribution and harm of this fungus was added in the introduction.
There are some imprecisions and inconsistencies between through the manuscript which must be corrected both in the abstract, results and discussion section and the conclusions which, otherwise, are not totally supported by data.
Response: We revised it.
Concerning the experimental plan, it is not clear how authors standardised the inoculum size and which inoculum rates were in fact used. This should be better described. In this view it is also unclear why mycelia instead of spores were used for the various tests.
Response: Mycelium blocks was used to inoculate into fresh walnuts. Spore suspension was not used. Because the spores grow slowly and need to be suspended in water. The inoculation hole will be humid and easy to rot after inoculation with spore suspension.
Below are some specific comments.
Reference is a bit old and it should be replaced by a more recent one.
L 35-37 Please change the sentence as “… nutrition and quantity, and safety issues related to the production of toxic compounds”
Response: We revised it.
L 37 Please delete “pathogenic”
Response: We revised it.
L 37 – 40 Please rephrase the sentence as it is grammatically incorrect, and its end is too vague, i.e. “et al” does not give any information to the readers.
Response: We revised it.
L 40-42 Please correct the sentence as “ .. while certain types of fungi usually produce mycotoxin during the pre-harvest periods.”
Response: We revised it.
L 43-46 Please correct the sentence as “ and highly important for fruits security and income generation”.
Response: We revised it.
L 47-51 The description provided into brackets for chemical preservatives does not provide any additional information for readers, but it is simply a repetition of the idea of using chemicals. Authors should change it by detailing the main content.
Response: We revised the instruction, and add more information about biological control.
L 54-56 Please delete the sentence as it is scientifically and practically incorrect. In its present form, the message delivered to readers is that “yeasts or bacteria used/proposed as biocontrol agents against fruit disease agent are considered harmless to humans”, also considering that biological agents undergo the Qualified Presumption of Safety which was developed to provide a pre-evaluation of the safety by EFSA. Moreover, the list of QPS recommended biological agents has recently been updated by EFSA in 2021.
Response: we deleted it.
L 56-57. Please correct “have” into “has”.
Response: We revised it.
L 57 -60 Please correct the sentence as it contains some errors (e.g. “pace”; should it be “Space”?), unclear/missing parts (“and forming formation”) and its end is too vague, i.e. “et al” does not give any information to the readers.
Response: We revised it.
L 61 -63 Please correct as “Application of biological agent can effectively contribute to…”
Response: We revised it.
L 71 Please correct “In order”
Response: We revised it.
Materials and Methods
L 86-87. Please indicate when walnuts were collected, how long and how, i.e. under which conditions they were stored before experiments were carried out.
Response: Fruits with disease symptoms were collected and the collecting time of walnuts was added.
L 86-87 Is “wukangxuan” a place or what else? Please clarify and detail better for clarity to the readers.
Response: “wukangxuan” is a place.
L 120-129 Authors should better detail how experiments were standardised and carried out. For example, no information is provided for the cultivation conditions nor for the standardization of the inocula: how authors can assure they used the same inoculum rates for the various fruits since “mycelium blocks” were used to inoculate the fruits?
Response: More information about methods was given. Each experiment was done in triplicate. The fungi were cultured on PDA medium for 2-3 days, and cut into a diameter of 5 mm mycelium blocks when the mycelium was about 2mm.
Also, how many walnut fruits were used to test the pathogenicity of the fungal strains on walnuts? Formula should be better explained by detailing what they differ for, what the representative number correspond to, and the difference between number of fruit disease and total fruit of disease.
Response: These formulas have been modified according to the comments of reviewers.
L 131-134 It is unclear what “PDA plants” correspond to. Please clarify by adding more details to this section.
L 132-134 Please add also 40°C to harmonise with results presented in figure 3c.
Response: We revised it.
L 131-137 According to L 137, assessment of the effects of different light/dark conditions, different temperature and pH values was made 2 days after inoculation. However, the incubation time is reported to be 24 hours. Such an inconsistency must be corrected.
Response: We revised it. The light condition refers to the time of light every day. "2 days" refers to the total inoculation time.
L 139 I suggest authors to change the heading of the section 2.5 as no evaluation of the presence of proteins was in fact made. Similarly, the heading of the section 2.5.2 is incorrect as it does not reflect the content of the section which in fact refers to the effects of different contents of supernatants added to agarized medium on the growth extent of the fungi.
Response: We revised it.
L 140-143 Authors should include an explanation for the choice to cultivate the Bacillus and Hanseniaspora strains for 14 hours as such a cultivation time is rather short, especially for the yeast. Also additional information on their inoculum rates and final cell loads after their growth, and incubation temperatures as well, should be included.
Response: These bacteria were in stationary phase after 14 hours incubation, the cell concentration reached an OD600 of 2.0. So Bacillus and Hanseniaspora strains for 14 hours were used for biological control experiments.
L 153-173 According to the description of the section 2.5.3, no cell free supernatant of the Bacillus/Hanseniaspora cultures was added to the liquid medium during the cultivation of the fungal strains. It is therefore unclear how antifungal activity was evaluated. Also the reference to Ye et al. [18] is not relevant since a different test was used by Ye et al compared to the present work to check for the antifungal activity (i.e. measurements of the inhibition zones on PDA plates by Ye et al. vs weight of the dried biomass following cultivation in liquid medium in the current study).
Moreover, it is important that authors include key information related to the preparation of the inoculum for the fungal strains and the corresponding inoculum rates.
Response: We modified the section 2.5.3. About 0.1g mycelium of pathogen was inoculated into 160 mL PDB culture solution. The supernatant of the Bacillus/Hanseniaspora cultures was added to the liquid medium after 24-hours incubation. The cultures were continued to incubate for 8,16 or 24 hours for antagonistic activity determination.
Results
Concerning the first part of this work, i.e. the isolation and characterization of fungal isolates, I do not agree with the authors’ decision to skip description of the corresponding results. Authors should therefore include the accession numbers of the isolates and the corresponding genus and species. These results are extremely relevant also considering that one of the stated aims of the work was to identify the main fungal pathogen of walnuts. Moreover, also the results of the pathogenicity tests of the fungal isolates on walnuts must be presented since this part of the experimental work is an important phase which resulted in the selection of the strongest pathogen.
Response: The colony characteristics and pathogenicity of 10 isolated fungal pathogens were added in figure 1. Identification results of fungal pathogens were added in table 1.
L 197-199 This part is incorrect and inconsistent with the description given in the M&M (section 2.4.1). First of all it is unclear why bacterial cell suspension was inoculated into fresh walnuts to confirm the pathogenicity of this isolate. Secondly, according to the M&M fungal mycelium was used in this test, while spore solution is reported to be used at L 198-199.
Response: In pathogenicity experiments, a piece of mycelium blocks with a diameter similar to the wound surface was used to inoculate into fresh walnuts. Spore suspension was not used. Because the spores grow slowly and need to be suspended in water. The inoculation hole will be humid and easy to rot after inoculation with spore suspension.
Table 1 – Is it correct that data recorded for the disease are expressed in mm? It seems to me that 0.8 mm is really a low value; how could authors have measured such values? Also data of the control uninoculated samples should be included in the table. Simply reporting that “The fruit inoculated with spore solution was more easily decayed than the control” (L 198-199) is rather vague for readers.
Response: The unit of diameter is wrong. It's not mm but cm. We have modified it.
Figure 3a; L 208-211 Please harmonise labelling of light conditions used for cultivation in fig. 3a with the description of the M&M section. In fact 24-h cultivation under dark is significantly different from heavy shading, and similarly 12 h light followed by 12 h dark is not the same as light shading. Such inconsistency must be removed and authors must clarify which conditions were in fact used since results showed that light conditions strongly affect the extent of fungal growth
Response: We have modified the figure.
L 208-209 The sentence must be revised as the verb is missing.
Response: We revised it.
L 219-220 The word “Different” must be written in lowercase letter.
Response: We revised it.
L 245 Please correct “insignificant” as “not significantly different”
Response: We revised it.
L 258-260 Please correct “significance” into “significantly”
Response: We revised it.
Please correct “untreatment” through the whole manuscript with “untreated” or “control”
Response: We revised it.
Figure 8 and the corresponding text at L 267-274 are not necessary as the do not provide any additional information than those in figures 4-7, and therefore they should be deleted.
Response: Figure 8 was deleted.
L 282-284 – Please edit the sentence as follows “This result suggests that antifungal activities of biocontrol agents could be attributed to their ability to damage fungal cells and antioxidant system, especially nucleic acid leakage of Bot. dothidea HT01”
Response: We revised it.
L 295-296 Please correct as “ … found that Aspergillus, Fusarium, …………..frequently isolated from African walnuts.”
Response: We revised it.
L 307 Please correct as “biocontrol agents”
Response: We revised it.
L 310-312 See my previous comments on “et al”.
Response: We revised it.
L 320-322 The sentence should be corrected as in fact the biocontrol agents did not inhibit, but promoted the nucleic acid leakage and malondialdehyde content.
Response: We revised it.

Reviewer 2 Report
General comments
Please check the References in-text and end-list for uniformity in style.
The authors should elaborate more on their findings compared to other studies, to their importance.
GenBank sequence accession number for your isolates requested
Please include some latest research findings, updated reviews in introduction and discussion part related to the topic.
Figures is not in printable quality. Also, some portions of the texts are losing their readability while sizing the image as per text area. Kindly provide better quality figure.
Make corrections in typographical errors as much as possible,
Many sentences do not have the correct punctuation and it is difficult to read the text.
English should be improved; grammar need for enhancement in many sentences and paragraphs.
The conclusion you have provided is quite brief and provide sufficient feedback on the main objectives of your study.
Author Response
Response to editor and reviewers
Dear Editor and Reviewers:
We thank you very much for your comments for our manuscript entitled “Exploring the efficacy of biocontrol bacteria against the fungal pathogen Botryosphaeria dothidea JNHT01 isolated from fresh walnut”. We have studied reviewer’s comments carefully and have made revision which marked in red in the revised manuscript. We invited a professional organization to revise the language of the article. We appreciate for Editors/Reviews’ warm work earnestly, and hope that the correction will meet with approval.
Looking forward to hearing from you.
Thank you and best regards.
Yours sincerely.
Hongmei Xiao
Reviewer #2:
Please check the References in-text and end-list for uniformity in style.
Response: We revised it.
The authors should elaborate more on their findings compared to other studies, to their importance.
Response: We revised through the whole manuscript, modified results and discussion.
GenBank sequence accession number for your isolates requested
Response: Identification results of fungal pathogens were added in table 1.
Please include some latest research findings, updated reviews in introduction and discussion part related to the topic.
Response: We updated references.
Figures is not in printable quality. Also, some portions of the texts are losing their readability while sizing the image as per text area. Kindly provide better quality figure.
Response: Figures have been modified to increase quality according to the comments of reviewers.
Make corrections in typographical errors as much as possible,Many sentences do not have the correct punctuation and it is difficult to read the text.
Response: We revised it.
English should be improved; grammar need for enhancement in many sentences and paragraphs.
Response: We revised it.
The conclusion you have provided is quite brief and provide sufficient feedback on the main objectives of your study.
Response: We revised it.
